# Facile Synthesis of Ultrastable Fluorescent Copper Nanoclusters and Their Cellular Imaging Application

**DOI:** 10.3390/nano10091678

**Published:** 2020-08-26

**Authors:** Wei Yan, Jianqiao Zhang, Muhammad Abbas, Yulian Li, Syed Zajif Hussain, Shazia Mumtaz, Zhengwei Song, Irshad Hussain, Bien Tan

**Affiliations:** 1Ministry-of-Education Key Laboratory for the Green Preparation and Application of Functional Materials, Hubei Collaborative Innovation Center for Advanced Organic Chemical Materials, Hubei Key Laboratory of Polymer Materials, Faculty of Materials Science and Engineering, Hubei University, Wuhan 430062, China; willieyancn2003@foxmail.com (W.Y.); songkukus@163.com (Z.S.); 2Key Laboratory of Material Chemistry for Energy Conversion and Storage, Ministry of Education, School of Chemistry and Chemical Engineering, Huazhong University of Science and Technology, Wuhan 430074, China; jqzhang1991@163.com (J.Z.); yulianli1993@163.com (Y.L.); 3Department of Chemistry & Chemical Engineering, SBA School of Science & Engineering, Lahore University of Management Sciences(LUMS), Lahore 54792, Pakistan; Abbas@UTDallas.edu (M.A.); syed.hussain@lums.edu.pk (S.Z.H.); shaziamumtaz5@gmail.com (S.M.)

**Keywords:** copper nanoclusters, glutathione, cellular imaging, fluorescence

## Abstract

Copper nanoclusters (Cu NCs) are generally formed by several to dozens of atoms. Because of wide range of raw materials and cheap prices, Cu NCs have attracted scientists’ special attention. However, Cu NCs tend to undergo oxidation easily. Thus, there is a dire need to develop a synthetic protocol for preparing fluorescent Cu NCs with high QY and better stability. Herein, we report a one-step method for preparing stable blue-green fluorescent copper nanoclusters using glutathione (GSH) as both a reducing agent and a stabilizing agent. High-resolution transmission electron microscopy (HRTEM), X-ray photoelectron spectroscopy (XPS) and electrospray ionization mass spectrometer (ESI-MS) were used to characterize the resulting Cu NCs. The as-prepared Cu NCs@GSH possess an ultrasmall size (2.3 ± 0.4 nm), blue-green fluorescence with decent quantum yield (6.2%) and good stability. MTT results clearly suggest that the Cu NCs@GSH are biocompatible. After incubated with EB-labeled HEK293T cells, the Cu NCs mainly accumulated in nuclei of the cells, suggesting that the as-prepared Cu NCs could potentially be used as the fluorescent probe for applications in cellular imaging.

## 1. Introduction

Due to their unique physicochemical properties, metal nanoclusters (M NCs) possess promising applications in the field of catalysis [1,2,3], sensors [4,5,6,7], photonics [8,9] and bioimaging [10,11,12,13]. With the size of nanoparticles decreasing down to the Fermi wavelength of an electron, the ultrafine particles are envisioned to bridge the “missing link” between metal atoms (showing special optical properties) and nanoparticles (showing surface Plasmon resonance), displaying molecule-like properties such as the strong fluorescence [14,15]. Compared with quantum dots (large size and high toxicity), lanthanide-doped nanomaterials (low luminescence efficiency) and organic dyes (poor photostability) [16,17], fluorescent nanoclusters exhibit many advantages including ultrasmall size, good photostability, low toxicity, good biocompatibility, which make them more attractive in the biological field [18,19,20]. Up to date, fluorescent gold and silver nanoclusters were widely explored while copper nanoclusters (Cu NCs) are much less explored mainly because of the difficulty in obtaining ultra-small size Cu NCs with good stability [2,21,22,23]. However, the great potential advantages over other nanoclusters including abundant available resources and comparable low cost intrigued more research interest. Chemical reduction and the ligand etching method are generally used for the preparation of Cu NCs [23,24,25,26,27], but they suffer from obvious drawbacks including complicated and time-consuming process, low quantum yield (QY) and poor stability. Using 4-aminothiophenol (PATP) as the thiol capping ligands and reducing agent, Feng et al. prepared Cu NCs. The QY of the obtained Cu NCs is as high as 24.6%, but the lifetime is only 14 days [17]. Ouyang et al. succeeded in synthesizing fluorescent Cu NCs using DNA as template. The QY is as high as 12%, however the fluorescence intensity begin to decrease just after 165 min [28]. Therefore, there is a dire need to develop a synthetic protocol for preparing fluorescent Cu NCs with high QY and better stability that can easily be upscaled. Many teams carried out research on Cu NCs in the bio-system to solve this problem [29,30,31,32,33,34]. J.C. Hao et al. induce the glutathione-capped CuNCs (GSH–CuNCs) and formed ordered assemblies, resulting in enhanced fluorescent properties [29]. L.Y. Lin used glutathione-capped CuNCs (with QY 1.3%) to detect Zn^2+^ basing on the aggregation-induced emission enhancement of GSH-capped Cu NCs [30]. However, both methods used in these literature synthesized CuNCs at room temperature with short-time heating. Although the synthesized CuNCs have close to red fluorescence, the low quantum fluorescence efficiency and poor stability limit their further application.

Herein, a simple one-pot thermo-reduced method is designed to prepare fluorescent Cu NCs employing glutathione (GSH) as a reducing agent and a protecting agent. In a typical experiment, GSH was first mixed with CuCl_2_, then NaOH solution was added, after stirring for 24 h, the solution was cooled down to room temperature. The resulting concentrated Cu NCs@GSH were precipitated by addition of isopropanol and after three purification cycles, and finally the Cu NCs@GSH were dispersed in water for further application. The effect of various synthetic parameters including the concentration of GSH, NaOH, reaction temperature and the reaction time on the fluorescence and stability of the Cu NCs were examined as optimization. The resulting Cu NCs were thoroughly characterized by high-resolution transmission electron microscopy (HRTEM), X-ray photoelectron spectroscopy (XPS) and electrospray ionization mass spectrometer (ESI-MS). The as-synthesized Cu NCs exhibited high QY (6.2%) and excellent stability (ion-stability, antioxidation stability, photostability and time-stability) and were successfully used for the imaging of HEK293T cells.

## 2. Materials and Methods

### 2.1. Reagents and Chemicals

All the chemical reagents used are at least of analytical grade. Anhydrous copper chloride (CuCl_2_) and glutathione (reduced) (GSH, 98%) were purchased from Sigma-Aldrich (Shanghai, China). Sodium hydroxide (NaOH, 99.5%) was bought from National Medicines Corporation, Ltd. of P.R. (Shanghai, China). The HeLa cells and human embryonic kidney cells HEK293T were purchased from the Shanghai Institute of Cells (Shanghai, China). Dimethyl sulfoxide (DMSO, 99%), 3-(4,5-dimethyl-2-thiazol-2-yl)-2,5-diphenyltetrazolium bromide (MTT, 98%) and ethidium bromide (EB) were obtained from J&K Chemicals (Shanghai, China). Dulbecco’s modified Eagle’s medium (DMEM) was purchased from Gibco (Shanghai, China). The ultrapure water (≥18 MΩ) obtained from a Milli-Q water purification system (Darmstadt, German) was used throughout.

### 2.2. Characterization

The fluorescence spectra were measured with an Enspire Multimode Plate Reader (PerkinElmer, Shanghai, China). The optical adsorption spectra were recorded on an UV2550 Spectrophotometer (SHIMADZU, Shanghai, China). High-resolution transmission electron microscopy (HRTEM) images were obtained with a Tecnai G^2^ F30 (FEI, Shanghai, China) at an accelerating voltage of 200 kV. The X-ray photoelectron spectroscopy (XPS) were measured by AXIS-ULTRA DLD (Shimadzu, Shanghai, China). The mass spectra were obtained with a micro TOF-Q electrospray ionization mass spectrometer (ESI-MS) (Bruker Daltonics, Shanghai, China). The optical density (OD) of the mixture was measured at 490 nm with a microplate absorbance reader (VersaMax (Molecular Devices), Shanghai, China). Cellular imaging was performed on a Confocal microscope (Olympus IX 81 + FV1000, Shanghai, China) using an excitation wavelength of 380 nm. All the experiments were conducted at room temperature, if not stated otherwise.

### 2.3. Synthesis of Blue-Green-Emitting Cu NCs@GSH

The GSH-stabilized CuNCs were prepared by a thermo-reduced method. In a typical experiment, GSH (1 mL, 9 mM) was mixed with CuCl_2_ (8.9 mL, 50 mM), and the solution became cloudy. Next, NaOH solution (100 μL, 1 M) was added dropwise with vigorous stirring until the mixture solution changed to be clear. Then the mixture was kept stirring at 80 °C for 24 h until the color changed from pale blue to purple. The purple solution of Cu NCs was then gradually cooled down to room temperature and stored in refrigerator at 4 °C. The resulting concentrated Cu NCs@GSH were precipitated by addition of isopropanol, collected through centrifugal at 10,000 rpm, and then redispersed in water. After 3 purification cycles, and finally the Cu NCs@GSH were dispersed in 10 mL water for further application.

### 2.4. Cell Culture and Cytotoxicity Assay

HeLa cells were cultured in Dulbecco’s modified Eagle’s medium (DMEM) including high glucose supplemented with 10% fetal bovine serum (FBS), 100 U of penicillin and 100-μg/mL streptomycin at 5% CO_2_, 37 °C. The cell viability was determined by the MTT assay according to the manufacturer’s instructions. Briefly, the HeLa cells were seeded in a 96-well plate in cell medium overnight. After being incubated with different concentrations of Cu NCs@GSH (0, 10, 50, 100, 200 and 400 μg/mL) for 12 h, 10 μL of MTT solution (5 mg/mL) was added to each well. After 4 h of incubation, 100 μL of DMSO was added to each well. The CuNCs@GSH are synthesized in aqueous phase, whose solubility depends on the ligand GSH. Moreover, the solubility of GSH is 500 g/L at pH 8.0–8.5.

### 2.5. Cellular Imaging

The human embryonic kidney cells HEK293T were grown in DMEM supplemented with 10% FBS at 37 °C and 5% CO_2_ for 24 h. The cells were then washed with phosphate-buffered saline (PBS), followed by incubation with Cu NCs (100 μL, 250 μg/mL) for 2 h at 37 °C and then washed with PBS three times, followed by incubation with EB solution (100 μL, 2 μg/mL) for 15 min. Cellular imaging was then performed on a Confocal microscope (Olympus IX 81 + FV1000) using an excitation wavelength of 380 nm and 583 nm after the cells were washed with PBS three times.

## 3. Results

### 3.1. Synthesis of Cu NCs@GSH

The GSH-stabilized blue-green fluorescent Cu NCs were prepared according to Scheme 1. The as-synthesized Cu NCs exhibited decent quantum yield (QY = 6.2% with quinine sulfate in 1-M H_2_SO_4_ as a reference) and excellent stability. The GSH was chosen as both the scaffold and reducing agent to protect the clusters possibly due to the strong interaction of the Cu^2+^ and appropriately placed cysteine residues in GSH during the encapsulation process [35,36,37]. Due to the biocompatibility of GSH, GSH-functionalized Cu NCs were found to be suitable for bio-applications due to their bioactive surface. This synthetic method can be categorized as a green one because no toxic chemical/solvent was used in this synthesis process and the reduction was induced thermally in the presence of GSH to produce Cu NCs@GSH, which were then used for cellular imaging. It is worth noticing that the reaction can be easily upscaled to produce more than 750 mL Cu NCs in a single synthesis step with decent fluorescence emission (Appendix A), which demonstrates the cost-effective and large-scale production of fluorescent metal nanoclusters.

### 3.2. Optimization Results for Reaction Conditions

We examined the effect of various synthetic parameters to produce highly fluorescent and stable Cu NCs including the concentration of GSH, NaOH, reaction temperature and the reaction time. In the systematic optimization process, the volume of GSH (1 mL), CuCl_2_ (8.9 mL), NaOH (0.1 mL) were kept constant to keep the same total volume, i.e., 10 mL, for each optimization reaction, the variation parameter was concentration. Firstly, we examined the effect of concentration of GSH to produce stable and fluorescent Cu NCs. As shown in Figure 1A,B, it was observed that with an increase in the concentration of GSH, the fluorescence intensity of maximum emission peak at 497 nm increased sharply and then decreased slowly while the color of solution turned from brown to purple and finally to little green (the absorption spectra are shown in Appendix A), which may be due to the lower concentration of GSH that was not sufficient to protect Cu NCs well. At higher concentration of GSH, the nucleation of the Cu NCs was trapped by the capping reagent, which reduced the chance of collision and prevented the NCs from growing [38]. Hence, by controlling the amount of GSH, the blue-green (maximum emission wavelength at 497 nm) fluorescent Cu NCs@GSH can be obtained and the highest blue-green fluorescence intensity was achieved by using 9-mM GSH. The final concentration of GSH is 0.9 mM. NaOH was not only helpful to make the cloudy suspension of GSH and Cu^2+^ clear, but it can also enhance the reducing capability of –SH of GSH [24]. In Figure 1C, the highest fluorescence intensity of Cu NCs was achieved when the concentration of NaOH was 1 M. The final concentration of NaOH is 10 mM. The lower (0.25 M, 0.5 M) and higher concentrations (2 M, 4 M) of NaOH were not helpful to further improve fluorescence intensity of Cu NCs. We also investigated the effect of temperature and reaction time to obtain the most stable and highly fluorescent Cu NCs. Figure 1D shows the effect of temperature on the fluorescence intensity of Cu NCs at the prolonged reaction time. Figure 1D and Figure 2 also show that 80 °C is the optimum temperature for the synthesis of highly fluorescent Cu NCs. It may due to the fact that a high temperature is not only helpful to overcome the energy barrier of the reduction process, but is also helpful to enhance the reaction rate of both NCs growth and digestion, therefore facilitating the structure optimization of Cu NCs [1,39,40,41,42]. The reducing ability of GSH at lower/room temperature (25 °C) was too low to achieve a decent yield of Cu NCs due to the slow reaction kinetics. However, by increasing the reaction temperature to 90 °C resulted in the drop of fluorescence intensity, which may be attributed to the aggregation of the ultrasmall Cu NCs into larger Cu NPs at higher temperature.

### 3.3. Characterization of As-Prepared Cu NCs@GSH

The as-synthesized Cu NCs@GSH were well dispersed in water and emitted an intense blue-green fluorescence under 365 nm irritation (Figure 3D). Figure 3A shows that the maximum excitation wavelength and emission wavelength are located at 372 nm and 497 nm, respectively. Meanwhile, as the blank control, the pure Cu^2+^, pure GSH and their mixture show no fluorescence (Appendix A). In Figure 3B, there is an obvious absorption peak around 600 nm, which may be attributed to the aggregation of Cu NCs@GSH (in Appendix A) and not the formation of large size Cu NPs. The location of emission peak remained unchanged even upon various excitation wavelengths (Figure 3C), implying that it was real fluorescence and not light scattering [43,44,45] and the relatively uniform surface state [46,47]. After freeze-drying, the lavender powder was obtained, which showed no fluorescence by irradiating at 365 nm while the strong blue-green fluorescence of Cu NCs@GSH can be recovered after their redispersion in water (Appendix A). The QY of as-prepared Cu NCs@GSH in aqueous solution was calculated to 6.2% using quinine sulfate (QY = 0.54 in 1 M H_2_SO_4_) as the reference [48]. It was also observed that by dispersing the Cu NCs in ethanol aqueous mixture, with increasing volume ratio of ethanol to water, the fluorescence intensity was gradually enhanced by increasing ethanol content (Appendix A), indicating that the Cu NCs@GSH exhibit the aggregation-induced emission enhancement (AIEE), which agrees with previous reports [49,50,51,52]. More important, the as-synthesized Cu NCs@GSH exhibited excellent stability (Figure 4).

As shown in Figure 4A, the fluorescence intensity of Cu NCs remained the same even in the solution with high ionic strength. The Cu NCs also exhibited good resistance to oxidation for that the addition of H_2_O_2_ influenced the fluorescence intensity very slightly (Figure 4B). After exposure to 372-nm excitation light for 7000 s, the fluorescence intensity of Cu NCs decreased by 13.4%, exhibiting better photostability than that organic dye Rhodamine 6G (Figure 4C). The storage under 4 °C for 3 months and 6 months reduced the fluorescence intensity of Cu NCs only by 5.5% and 16.5%, respectively, exhibiting their decent stability over a long period of time that would be very beneficial for their potential bio-applications (Figure 4D).

The FTIR spectrum shows a typical absorption band of the carboxyl group at 1712 cm^−1^ that corresponds to the weak interaction of the glycine residues of GSH with Cu NCs (Figure 5A). Compared with the pure GSH alone, the characteristic peak of SH at 2525 cm^−1^, disappeared in the GSH-stabilized Cu NCs, suggesting the interaction between thiol group and Cu NCs [49]. XPS analysis showed that the binding energy of Cu 2p_3/2_ and Cu 2p_1/2_ were located at 932.9 eV and 952.4 eV (Figure 5B) indicating that Cu NCs are indeed composed of both Cu(0) and Cu(I) species, and the existence of Cu(I) also supports their improved stability and quantum yield [23,24]. Furthermore, there was no peak displayed around 942.0 eV, demonstrating the absence of Cu(II) in Cu NCs [52]. In addition, the S 2p3/2 peak located at 166.3 eV confirmed the covalent interaction of Cu NCs with the SH group (Figure 5E) [53]. The peaks of S(1s), C(1s), N(1s), Na, O(1s), Cu (2p), and Na(1s) are clearly visible in the spectrum (Figure 5F), further indicating that Cu NCs are stabilized by GSH [54]. The exact atomic composition of Cu NCs was determined using ESI-MS analysis. From the MS spectrum (Figure 5C), the highest peak located at 1942.7 could be attributed to the structure composition of [Cu_11_(GSH)_4_+Na+8H]. The HRTEM image (Figure 5D) clearly showed the formation of uniform Cu NCs with an average diameter of 2.3 ± 0.4 nm.

### 3.4. Cytotoxicity Assay and Cellular Imaging

Prior to the biologic applications, the cytotoxicity of the Cu NCs@GSH was studied by the MTT method. As shown in Figure 6, after different concentrations of Cu NCs@GSH were incubated with HeLa cells for 12 h, there was no obvious decrease of the cell viability, but the slight cell growth. These results clearly suggest that the Cu NCs@GSH are biocompatible and suitable for potential bioimaging applications. In order to explore the role of Cu NCs in the field of fluorescent bioimaging, we chose human embryonic kidney cells HEK293T in this study (Figure 7). After incubation with Cu NCs, all the cells showed clear cells morphology and a bright blue fluorescence from the intracellular region (Figure 7A,B), while no fluorescence was observed from the control experiment under the similar conditions (Appendix A). EB is a commonly used organic dye in tissue labeling due to its specific binding to nuclei. Comparing the images of EB-labeled HEK293T cells (Figure 7C) and the overlap of the above three images (Figure 7D), it can be seen that the Cu NCs were also mainly accumulated in nuclei of the cells, suggesting that the as-prepared Cu NCs could potentially be used as the fluorescent probe for applications in cellular imaging.

## 4. Conclusions

In summary, a facile synthetic protocol is developed for the preparation of highly fluorescent Cu NCs using GSH as both a reducing and stabilizing agent, which can be upscaled easily. The as-synthesized Cu NCs exhibit bright blue-green fluorescence, high QY and excellent stability. Moreover, the Cu NCs exhibit low cytotoxicity and, therefore, possess great potential for cellular imaging to study various biologic processes.

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
