# Peer review of "Facile Synthesis of Ultrastable Fluorescent Copper Nanoclusters and Their Cellular Imaging Application"

_nanomaterials, 2020, doi:10.3390/nano10091678_

Round 1
Reviewer 1 Report
The authors described a one-step method for preparing blue-green fluorescent copper nanoclusters (Cu NCs) with high QY and good stability. using glutathione (GSH) as both a reducing agent and a stabilizing agent. High-resolution transmission electron microscopy, X-ray photoelectron spectroscopy and electrospray ionization mass spectrometer were employed for characterization. Cu NCs GSH are biocompatible and could potentially be used as the fluorescent probe for applications in cellular imaging
The revision of the manuscript is required before acceptance and my comments are below
- The synthetic pattern lacks of novelty: novelty of the work must be discussed in the Introduction and inherent recent literature about similar bio-systems must be added.
- In Sect. 3.1 some recent references on AIE fluorescent systems (lines 179-183) with well-studied optical properties should be added, as for example: Crystals (2020), 10(4), 269.
- The spectroscopic data collected on the complexes must be improved. QYs are really too poor for every kind of applications. Please, discuss this issue at least.
- Given that water solution is required for applications, the solubility in pure water (without DMSO) of the adduct must be given
- English in the manuscript must be severely improved, a general detailed revision is to be recommended due to typos and grammar mistakes
The work actually provides results in the field of nanoclusters for bio-imaging but can be considered for publication on Nanomaterials only after major revision.
Author Response
- The synthetic pattern lacks of novelty: novelty of the work must be discussed in the Introduction and inherent recent literature about similar bio-systems must be added.
Response: Thanks for the instructive comments. Inherent recent literature about similar bio-systems were added. from line 58 to line 62, the novelty of the work were also added from line 62 to line 65 in the Introduction part as following:
Many teams carried out research on Cu NCs in the bio-system to solve this problem.[1-6] .J.C. Hao et al. induce the glutathione-capped CuNCs (GSH-CuNCs) and formed ordered assemblies, resulting in enhanced fluorescent properties[1]. L.Y. Lin used glutathione-capped CuNCs (with QY 1.3%) to detect Zn2+ basing on the aggregation induced emission enhancement of GSH-capped Cu NCs[2]. Howerver, both methods used in these literature synthesized CuNCs at room temperature with short-time heating. Although the synthesized CuNCs have close to red fluorescence, the low quantum fluorescence efficiency and poor stability limit their further application.
- In Sect. 3.1 some recent references on AIE fluorescent systems (lines 179-183) with well-studied optical properties should be added, as for example: Crystals (2020), 10(4), 269.
Response: Thanks for the suggestion. Some recent references on AIE fluorescent systems (original lines 179-183,now lines 186-190) with well-studied optical properties were added, as following:
44.Jeong, M.; Kim, H.M.; Lee, J.S.; Choi, J.H.; Jang, D.S. (-)-Asarinin from the Roots of Asarum sieboldii Induces Apoptotic Cell Death via Caspase Activation in Human Ovarian Cancer Cells. Molecules 2018, 23, 1849.
45.Diana, R.; Caruso, U.; Di Costanzo, L.; Bakayoko, G.; Panunzi, B. A Novel DR/NIR T-Shaped AIEgen: Synthesis and X-Ray Crystal Structure Study. Crystals 2020, 10.
- The spectroscopic data collected on the complexes must be improved. QYs are really too poor for every kind of applications. Please, discuss this issue at least.
Response: Thanks for your instructive comments and suggestion. There are many teams carrying out research on glutathione protected Cu NCs.[1-6] However, none of the QYs in these literatures is higher than that of ours. That’s the reason why we developed our own method at higher temperature for longer time to ensure the copper nanoclusters with improving QYs.
We had just proved the prepared copper nanoclusters to be potential for cell imaging, and the QY is enough for the experiment described in our paper. But the COVID- 19 makes it very hard to make more bio-expeiments. We will try to develop some more experiments in the future. (for more details,please see anwer to reviwer 1,point 3)
[1] J. Yuan, L. Wang, Y. Wang. Stimuli‐Responsive Fluorescent Nanoswitches: Solvent‐Induced Emission Enhancement of Copper Nanoclusters. Chemistry – A European Journal, 2020, 26, 3545-3554.
[2] L. Lin, Y. Hu, L. Zhang. Photoluminescence light-up detection of zinc ion and imaging in living cells based on the aggregation induced emission enhancement of glutathione-capped copper nanoclusters. Biosens Bioelectron, 2017, 94, 523-529.
[3] A. Baghdasaryan, R. Grillo, S. Roy Bhattacharya. Facile Synthesis, Size-Separation, Characterization, and Antimicrobial Properties of Thiolated Copper Clusters. ACS Applied Nano Materials, 2018, 1, 4258-4267.
[4] S. Chen, Z. HuangandQ. Jia. Electrostatically confined in-situ preparation of stable glutathione-capped copper nanoclusters for fluorescence detection of lysozyme. Sensors and Actuators B: Chemical, 2020, 319,
[5] R. JaliliandA. Khataee. Aluminum(III) triggered aggregation-induced emission of glutathione-capped copper nanoclusters as a fluorescent probe for creatinine. Mikrochim Acta, 2018, 186, 29.
[6] F. Qu, Q. Yang, B. Wang. Aggregation-induced emission of copper nanoclusters triggered by synergistic effect of dual metal ions and the application in the detection of H2O2 and related biomolecules. Talanta, 2020, 207, 120289.
- Given that water solution is required for applications, the solubility in pure water (without DMSO) of the adduct must be given
Response: Thanks for your comments. The CuNCs@GSH are synthesized in aqueous phase, whose solubility depends on the ligand GSH. And the solubility of GSH is 500 g/L at pH 8.0~8.5.
- English in the manuscript must be severely improved, a general detailed revision is to be recommended due to typos and grammar mistakes
Response: We appreciate the reviewer’s thoughtful comment and suggestion. Extensive editing of English language and style were carried out for the whole original manuscript

Reviewer 2 Report
Wei Yan et al. described a one-pot method to prepare fluorescent Cu NCs using glutathione (GSH) as a reducing agent in this manuscript. Bright blue-green emission, high photostability and biocompatibility were reported as main advantages of the nanoparticles, which are prerequisites for cellular imaging applications. The MS contains interesting results. However, several points should be addressed before accept.
- Y-axis should be adjusted (for example from 0 to 1000) in order to see magnitude of fluorescent peaks at 427 nm (Fig S2).
- Figures S2 and S3 are not described in the main text of the MS.
- It is not clear what authors tried to explain with red circles (Fig S3).
- Due to the reason of same content on Fig S5A and Fig S5B, I suggest to cut Fig S5A.
- Fluorescence intensity decrease may be a reason of aggregation in time (Fig 4D). Did authors use sonication before each measurement?
- Containing two amino groups, ethidium bromide (EB) has strong affinity to copper. It means the presence of each component could affect each other resulting nuclei targeting concluded in the paper. I would suggest using higher magnification on CLSM and utilize other nuclei dye(unfortunately many of them contain amino groups) or membrane specific dye to show localization of Cu NCs.
- The difference in color of emission can easily be recognized between Fig 3D Fig 7B (blue and blue-green emission). Can the authors explain this difference?
- Please support the statement “while the color of solution turned from brown to purple and finally to little green” (lines 135, 136) with corresponding absorption spectra.

Author Response
- Y-axis should be adjusted (for example from 0 to 1000) in order to see magnitude of fluorescent peaks at 427 nm (Fig S2).
Response: Thanks for your suggestion. The picture is adjusted as follow:
Original picture in Supporting Information:
The new picture in Supporting Information:
Fig S2. Fluorescence emission spectra of GSH, Cu2+ solution and mixture of GSH and Cu2+.
- Figures S2 and S3 are not described in the main text of the MS.
Response: Thanks for your comments. The discussion of Figure S2 is added in the paper.
The as-synthesized Cu NCs@GSH were well dispersed in water and emitted an intense blue-green fluorescence under 365 nm irritation (Fig 3D). Fig 3A shows that the maximum excitation wavelength and emission wavelength are located at 372 nm and 497 nm respectively. Meanwhile, as the blank control, the pure Cu2+, pure GSH and their mixture show no fluorescence (Fig S2). In Fig 3B, there is an obvious absorption peak around 600 nm, which may be attributed to the aggregation of Cu NCs@GSH (in Fig S3), and not the formation of large size Cu NPs.
- It is not clear what authors tried to explain with red circles (Fig S3).
Response: Thanks for your comments. The explain of figure S3 is is added in the paper.
The as-synthesized Cu NCs@GSH were well dispersed in water and emitted an intense blue-green fluorescence under 365 nm irritation (Fig 3D). Fig 3A shows that the maximum excitation wavelength and emission wavelength are located at 372 nm and 497 nm respectively. Meanwhile, as the blank control, the pure Cu2+, pure GSH and their mixture show no fluorescence (Fig S2). In Fig 3B, there is an obvious absorption peak around 600 nm, which may be attributed to the aggregation of Cu NCs@GSH (in Fig S3), and not the formation of large size Cu NPs.
- Due to the reason of same content on Fig S5A and Fig S5B, I suggest to cut Fig S5A.
Response: Thanks for your suggestion. The picture is adjusted as follow:
Original picture of Fig5:
|
Fig S5. Histogram of Cu NCs@GSH in different vol% Ethanol : water mixture.
- Fluorescence intensity decrease may be a reason of aggregation in time (Fig 4D). Did authors use sonication before each measurement?
Response: Thanks for your question. Before the measurement, the solution has been dispersed using sonication.
- Containing two amino groups, ethidium bromide (EB) has strong affinity to copper. It means the presence of each component could affect each other resulting nuclei targeting concluded in the paper. I would suggest using higher magnification on CLSM and utilize other nuclei dye(unfortunately many of them contain amino groups) or membrane specific dye to show localization of Cu NCs.
Response: Thanks for your suggestion for the dye, that’s a very good question. Nowdays Ethidium bromide is commonly used in the imaging of noble metal nanoclusters[1-4]. Tao et al. reported a ligand-induced etching process for preparing highly fluorescent PEI-templated gold nanoclusters (PEI-AuNCs), and they used Ethidium bromide to stain cells[1]. Yang et al. fabricated a fluorescence biosensing platform for human immunodeficiency virus gene (HIV-DNA) detection ,which was based on luminescent DNA-scaffolded silver nanoclusters (DNA/AgNCs) and autonomous exonuclease III (Exo III)-assisted recycling signal amplification. In this paper Ethidium bromide also was used[2]. As researchers pay more and more attention to the application of copper nanoclusters(Cu NCs) in cell imaging, Ethidium bromide was also used with copper nanoclusters. Ramadurai et al. reported the one-pot synthesis of 3-mercaptopropylsulfonate (MPS) protected Cu NCs where Ethidium bromide is one of the required materials[5]. Goswami et al. synthesized the Transferrin (Tf) templated luminescent blue copper nanoclusters (Tf-CuNCs). In this report, Ethidium bromide was used the products of PCR when Tf-CuNCs was existed. [6].After careful study of all the related reference which proofed that ethidium bromide would not interfere with the Cu NCs to bind to nuclei, we chose Ethidium bromide as the staining material in our system.
- Tao, Y.; Li, Z.; Ju, E.; Ren, J.; Qu, X. Polycations-functionalized water-soluble gold nanoclusters: a potential platform for simultaneous enhanced gene delivery and cell imaging. Nanoscale 2013, 5, 6154-6160.
- Yang, W.; Tian, J.; Wang, L.; Fu, S.; Huang, H.; Zhao, Y.; Zhao, S. A new label-free fluorescent sensor for human immunodeficiency virus detection based on exonuclease III-assisted quadratic recycling amplification and DNA-scaffolded silver nanoclusters. Analyst 2016, 141, 2998-3003.
- Zhang, J.; Li, C.; Zhi, X.; Ramon, G.A.; Liu, Y.; Zhang, C.; Pan, F.; Cui, D. Hairpin DNA-Templated Silver Nanoclusters as Novel Beacons in Strand Displacement Amplification for MicroRNA Detection. Anal Chem 2016, 88, 1294-1302.
- Zhang, X.; Liu, S.; Song, X.; Wang, H.; Wang, J.; Wang, Y.; Huang, J.; Yu, J. DNA three-way junction-actuated strand displacement for miRNA detection using a fluorescence light-up Ag nanocluster probe. Analyst 2019, 144, 3836-3842.
- Ramadurai, M.; Rajendran, G.; Bama, T.S.; Prabhu, P.; Kathiravan, K. Biocompatible thiolate protected copper nanoclusters for an efficient imaging of lung cancer cells. J Photochem Photobiol B 2020, 205, 111845.
- Goswami, U.; Dutta, A.; Raza, A.; Kandimalla, R.; Kalita, S.; Ghosh, S.S.; Chattopadhyay, A. Transferrin-Copper Nanocluster-Doxorubicin Nanoparticles as Targeted Theranostic Cancer Nanodrug. ACS Appl Mater Interfaces 2018, 10, 3282-3294.
As to the CLSM image with a higher magnification version, we had some more pictures here. Thanks very much for your suggestion.
- The difference in color of emission can easily be recognized between Fig 3D Fig 7B (blue and blue-green emission). Can the authors explain this difference?
Response: Thanks for your question. In Fig 3D, the solution shows blue and blue-green emission under UV light, which is the color observed by eyes. In Fig 7B, the different color is archived under the blue channel of confocal laser scanning microscopy, which is to explain the fluorescence from the intracellular region.
- Please support the statement “while the color of solution turned from brown to purple and finally to little green” (lines 135, 136) with corresponding absorption spectra.
Response: Thanks for your comment. The absorption spectra of solution with different colors is as follow. It can be seen that the peak around 600 nm becomes stronger and then weaker.

Reviewer 3 Report
The MS requires minor revisions for acceptance, for more details see the enclosed file.

Author Response
The MS requires minor revisions for acceptance, for more details see the enclosed file.
The MS 903643 introduces facile synthesis of Cu NCs and their careful characterization, thus, it deserves publishing in Nanomaterials, although the confocal images reveal rather small potential in cellular imaging. It is too early to claim “great potential” in bioimaging, nevertheless, it opens a door for further improvement. The MS should be revised before acceptance.
- The subdivisions “Synthesis …” and “Optimization…” should be combined. This will avoid the Duplications.
Response: Thanks very much for the suggestion. The subdivisions “Synthesis …” and “Optimization…” were added as red paragraph from“In a typical experiment”(line67) to “stability of the Cu NCs were examined as optimization”(line73)to avoid the Duplications as following:
Herein, a simple one-pot thermo-reduced method is designed to prepare fluorescent Cu NCs employing glutathione (GSH) as a reducing agent and a protecting agent. In a typical experiment, GSH was first mixed with CuCl2, then NaOH solution was added ,after stirring for 24 h the solution was cooled down to room temperature. The resulting concentrated Cu NCs@GSH were precipitated by addition of isopropanol and after 3 purification cycles, and finally the Cu NCs@GSH were dispersed in water for further application.The effect of various synthetic parameters including the concentration of GSH, NaOH, reaction temperature and the reaction time on the fluorescence and stability of the Cu NCs were examined as optimization.
- The authors should represent the synthetic conditions by concentrations of the components in the synthetic mixture, not those of the initial solutions. The concentrations of initial solutions should be replaced by the real concentrations both in discussion section and in Figure 1.
Response: Thanks for your suggestion. In order to discuss the effect of GSH concentration, the initial concentration has been used. For GSH, the real concentration should be the 10% of initial concentration.
- The reaction making the cloudy solution of Cu(SR)2 clear after the alkalization by NaOH must be specified.
Response: Thanks for your instructive comments and suggestion. When the copper precursor is mixed with GSH, the solution immediately becomes turbid milky white. This is because the sulfhydryl group (-SH) on GSH reacts with copper ions to form a hydrogel, and the addition of NaOH can make the hydrogel re-dissolve; The pH of (-SH) on GSH is 9.35. Adding NaOH to adjust the pH of the solution to about 10.00 can activate -SH, thereby enhancing its reducing ability. Therefore, the cloudy solution of Cu(SR)2 becomes clear after the alkalization by NaOH.
- The Scheme 1 is inconvenient, to improve it all reactions (both complex formation and redox) undergoing during the synthesis should be introduced.
Response: Thanks for your suggestion. The changed picture is shown as followed.
Original picture:
Scheme 1. The schematic illustration of thermo-reduced synthesis of fluorescent Cu NCs@GSH.
New picture:
Scheme 1. The schematic illustration of thermo-reduced synthesis of fluorescent Cu NCs@GSH.

Round 2
Reviewer 2 Report
Reviewer thanks authors for swift revision of the manuscript. All comments have been thoroughly addressed in revised form. Recommendation - accept.